# Evaluating the Effects of Controlled Drainage on Nitrogen Uptake, Utilization, Leaching, and Loss in Farmland Soil

**Xu Dou, Haibin Shi \*, Ruiping Li, Qingfeng Miao**  **, Jianwen Yan and Feng Tian**

College of Water Conservancy and Civil Engineering, Inner Mongolia Agricultural University, Huhhot 010018, China; nmgdx@imau.edu.cn (X.D.); nmglrp@imau.edu.cn (R.L.); imaumqf@imau.edu.cn (Q.M.); yanjianwen@imau.edu.cn (J.Y.); tianfeng@emails.imau.edu.cn (F.T.)

\* Correspondence: shb@imau.edu.cn

**Abstract:** Controlling drainage during the growth stage is one of the means to provide suitable water and fertilizer conditions for crops, alleviate environmental pollution, and increase crop yield. Therefore, in this study, we studied three drainage treatments: free drainage (FD) and growth-stage subsurface controlled drainage (CD) at depths of 40 cm (CWT1) and 70 cm (CWT2). We used the HYDRUS-2D model to simulate the dynamic changes of $NO_3$-N in the 0–100 cm soil layer as well as $NO_3$-N uptake by crops, leaching after irrigation and fertilization, and loss through subsurface pipes in 2020 (model calibration period) and 2021 (model validation period). The degree of agreement between the simulated and measured values was high, indicating a high simulation accuracy. CD increased the soil $NO_3$-N content and crop $NO_3$-N uptake, and decreased $NO_3$-N leaching and loss. We observed significant differences in the soil $NO_3$-N content after irrigation at the budding stage of oilseed sunflower between CD and FD, with the largest difference seen for the 0–40 cm soil layer. CD increased crop yield, and the average oilseed sunflower yield of the CWT1 and CWT2 treatments increased by 4.52% and 3.04% relative to the FD treatment ($p < 0.05$). CD also enhanced nitrogen use efficiency. In moderately salinized soil, CD at 40 cm (CWT1) reduced the nutrient difference in vertical and horizontal directions while retaining water and fertilizer. CWT1 stabilized the groundwater depth, reduced the hydraulic gradient of groundwater runoff, and decreased the drainage flow rate. The $NO_3$-N leaching and loss dropped, which promoted crop nitrogen uptake and utilization, improved nitrogen use efficiency, reduced nitrogen loss, and had a positive effect on protecting the soil and water environment. The results demonstrate that CD is a suitable drainage method for the experimental area.

**Keywords:** nitrate; controlled drainage; oilseed sunflower; HYDRUS-2D

## 1. Introduction

Nitrogen loss is an important factor that causes non-point-source pollution of farmland and deterioration of surface-water and groundwater quality, especially when residual chemical fertilizers due to excessive application of fertilizers enter the surrounding water bodies with drainage. This not only wastes fertilizers but also is the primary cause of water and soil pollution [1,2]. As an important means of farmland improvement and prevention of soil salinization, as well as reducing the groundwater level of farmland to prevent waterlogging, subsurface drainage is widely used in agricultural activities [3]. With an increasing awareness of water and soil environmental protection, however, this subsurface free drainage (FD) has been found to have a negative impact on surface water [4], causing soil and water pollution, resulting in eutrophication of river and lake water quality, and having a large impact on the environment [5]. In addition, subsurface FD can increase the output of $NO_3$-N in the soil profile. $NO_3$-N transport and its adverse effects on surface water quality have raised increasing concern [6]. Excessive use of nitrogen has caused a serious loss of $NO_3$-N in the drainage process [7], and this problem in farmland has

attracted significant attention to formulate scientific and reasonable protection measures to reduce the nutrient output from farmland drainage to receiving water bodies [8]. Therefore, the development of controlled drainage (CD) technology has good prospects.

CD is a new type of agricultural drainage management measure based on FD to periodically increase the elevation of the subsurface drain outlet [9]. The height of the drain outlet can be adjusted at different times of the year according to the crop growth stage or the field capacity requirements [10], thereby providing a good environment for crop growth and a favorable basis for field mechanical driving. In arid areas, CD technology is used to reduce drainage [7] and nutrient loss [11] and to alleviate water eutrophication and environmental pollution [12]. During the crop-growing season, CD can increase crop yield and nitrogen uptake, especially under drought conditions [13]. The concept of controlled drainage is to discharge the water precisely as needed, and discharge it when necessary [9]. Subsurface drainage system, however, are usually designed for continuous drainage without considering water and nutrient wastage and impacts on crop productivity [14]. CD is a management measure that achieves sustainable development of crop production on the basis of subsurface drainage, improves agricultural water management, and reduces environmental pollution caused by subsurface drainage without affecting crop yield and increasing costs [15].

As people pay greater attention to environmental pollution, CD technology has also been widely adopted worldwide [15,16]. CD is applied during the crop growth stage to improve water and nutrient use efficiencies and enhance crop yield, while FD is used during the non-growth stage to leach soil salts to prevent and improve salinized soils [17]. In the crop growth stage, the problem of excessive drainage of water and nutrients is tackled using CD technology [18] to reduce the pollution of water and soil environment, increase the soil water content by regulating groundwater level, increase soil nutrients, and provide suitable crop growth conditions. CD reduces drainage rates and volumes and reduces nutrient loss, which in turn reduces nutrient load and concentration in drains, thus playing a positive role in water conservation and environmental protection [19]. Wesstrom and Messing (2007) [20] have reported that plots installed with CD had reduced nitrogen loss relative to FD, but increased crop nitrogen uptake and yield, thus improving nitrogen use efficiency. Conversely, other studies have suggested little or no effect of CD on crop yield [5]. Therefore, to achieve the mutually beneficial goal of improving crop yield while protecting the environment, the implementation of CD needs to appropriately adjust the drainage depth, correct the subsurface drain spacing, and determine the appropriate drainage period according to different local characteristics.

The HYDRUS model has developed into a mature soil physical environment simulation tool that is widely used to simulate the changes of soil moisture [21,22], nutrients [23,24], salts [25,26], and temperature [27]. The HYDRUS model can capture the dynamic properties of these factors and be used to optimize their spatial and temporal distribution. Compared with DRAINMOD [16], SWAT [28], RZWQM2 [29], and other models, the HYDRUS-2D model not only simulate changes in $NO_3$-N content in soil and the loss of $NO_3$-N under CD conditions, but also can simulate crop $NO_3$-N uptake and utilization and the $NO_3$-N leaching process. The HYDRUS-2D model is widely used in Hetao irrigation area. This model can better simulate soil moisture, salts and nutrients, and provide theoretical basis for local farmland water saving, salt control and nutrient utilization. For example, Chen et al. (2020) [30] used HYDRUS-2D to analyze the soil nitrogen uptake and utilization as well as the leaching and soil nitrogen concentration changes in detail. That study showed that the HYDRUS-2D model performed well in simulating soil $NO_3$-N balance. In this study, we used the HYDRUS-2D model to calibrate and verify the nitrogen balance under CD conditions. Compared with Chen et al. (2020), we also analyzed the loss of soil nitrogen through subsurface drains. The results showed that the HYDRUS-2D model could systematically model the nitrogen budget under crop growth conditions and correctly calculate the soil $NO_3$-N balance process.

The Hetao Irrigation District of Inner Mongolia of China is an important grain production area but has an increasing salinization problem. While using subsurface drainage technology to improve and prevent soil salinization in the irrigation area, water and nutrient losses during the crop growth stage are not properly considered. Additionally, the soil salinization in the region is a serious problem, resulting in poor soil permeability. In the middle and late stages of oilseed sunflower growth, irrigation cannot be performed, or it will cause serious seedling death, which will lead to water and nutrient deficiencies in the late stage of oilseed sunflower growth. Therefore, in this study, we used CD technology to provide suitable water and nutrients for oilseed sunflowers in the late crop growth stage. We used the HYDRUS-2D model to investigate the effects of different drainage methods on the soil $NO_3$-N transport, crop $NO_3$-Nutilization, leaching of $NO_3$-N by irrigation and rainfall, and the loss of $NO_3$-N in moderately saline soil in the Hetao Irrigation District. The objective was to achieve the best water and nutrient supply, gain an in-depth understanding of the response pattern of drainage methods to the soil–crop system, and realize high efficiency of water and nitrogen use.

## 2. Materials and Methods

### 2.1. Study Tests

The test field is located in the subsurface drainage comprehensive test area the downstream of Hetao Irrigation District, Ulat Irrigation District, Bayannur, China, at latitude 40°45′28″ N, longitude 108°38′16″ E, and elevation 1018.88 m above sea level. The climate of the test area belongs to the middle temperate continental climate, with complex and variable temperature, windy and dry, sufficient sunshine, strong evaporation, and little precipitation. The multi-year average temperatures of the test area is 6~8 °C; the average precipitation is 196~215 mm, evapotranspiration is 2173 mm, and sunlight hours are 3231 h. It is a classic dry area. The main soil physicochemical properties of the study area are shown in Table 1. The average mean values of groundwater depth in 2020 and 2021 during the crop reproductive period were 1.2 m and 1.3 m. The soil basal fertility for organic matter, total nitrogen, effective nitrogen, effective phosphorus, and quick-acting potassium mass ratios were 13.54 g/kg, 0.85 g/kg, 86 mg/kg, 9.432 mg/kg, and 218 mg/kg, etc.

**Table 1.** Physical properties of soil in the experimental area.

| Soils Layer (cm) | Sand (%) | Silt (%) | Clay (%) | Soil Bulk Density (g·cm$^{-3}$) | Soil Field Capacity (cm$^3$·cm$^{-3}$) |
|---|---|---|---|---|---|
| 0–20 | 4.72 | 84.84 | 10.44 | 1.444 | 31.77 |
| 20–40 | 9.99 | 79.36 | 10.65 | 1.47 | 35.13 |
| 40–60 | 5.68 | 84.15 | 10.17 | 1.473 | 35.65 |
| 60–80 | 7.44 | 85.23 | 7.33 | 1.485 | 34.75 |
| 80–100 | 3.16 | 87.18 | 9.66 | 1.487 | 35.69 |
| 100–120 | 2.03 | 92.75 | 5.22 | 1.489 | 35.91 |

### 2.2. Test Design

The test was selected to be carried out in field plots, three treatments were set up, three replicates were set up in each plot, and there were nine plots in all. The depth of free drainage (FD) and controlled drainage was 40 cm (CWT1) and 70 cm (CWT2), etc., and the depth of drainage for spring irrigation was 100 cm in all cases (Figure 1). The experiment consisted of three treatments: free drainage to 100 cm (FD), controlled drainage to 40 cm (CWT1), and controlled drainage to 70 cm (CWT2). Drainage ports from the ground were 40 cm (No.1), 70 cm (No.2), and 100 cm (No.3) (Figure 1). We started the spring irrigation on 20 May 2020 and 23 May using an irrigation quota of 900 m$^3$/ha. All treatments had the same drainage to 100 cm by having ports No.1 and No.2 closed and drainage port No.3 open. Over the sunflower growth period, drainage events occurred in response to rainfall using drainage pipe setups, as shown in Figure 1. Drainage treatments were set up during the growth stage. For CWT1, port 1 was open, and 2 and 3 were closed. For

CWT2, port 2 was open, with 1 and 3 closed; for FD, 1 and 2 were closed, with 3 open. Each plot laid 2 subsurface pipes; the subsurface pipes had a buried depth of 100 cm, spacing of 20 m, pipe diameter of 80 mm, and slope of 0.1%. The test plots were 40 m long and 30 m wide, each test plot was spaced 10 m apart, and there was a protection zone around the plots, which was isolated by burying a 1 m deep polyvinyl chloride plastic sheet to prevent mutual interference. The test area was improved with laser leveling and saline soil in 2019, and desulfurization gypsum (30 t/hm$^2$) was added to the soil in the test area to displace the harmful sodium ions adsorbed by the soil; fine sand (dune sands) (85.05 m$^3$/hm$^2$) was applied to improve the permeability of the soil.

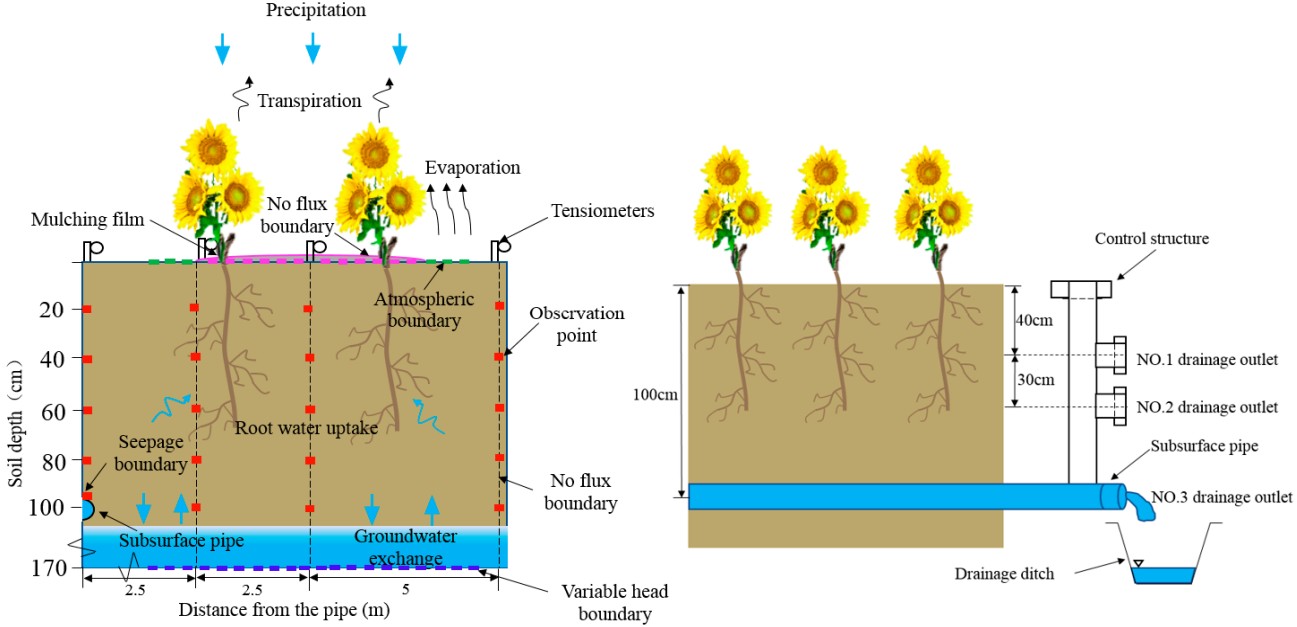

**Figure 1.** Field modeling area, boundary conditions, and control drainage devices.

We sowed the oilseed sunflowers (referred to as sunflower), variety "Ao33" on row distance of 60 cm and plant distance of 20 cm on 1 June 2020 and 4 June 2021 using 290 kg diammonium phosphate/ha (containing N 18%, P$_2$O$_5$ 44%) and 150 kg potassium sulfate/ha (containing K$_2$O 50%). We applied a post-seeding N application of 130 kg urea/ha during bud-breaking on 18 July 2020 (48 days after sowing, DAS48) and 26 July 2021 (DAS53). We harvested the sunflowers on 29 September 2020 and 3 October 2021 (DAS122). Immediately after fertilizer application, the mulch was covered with film for manual spot sowing and the hole was covered with fine sand after sowing, with a planting density of $4.95 \times 10^4$ plants/ha. The irrigation mode was frontier irrigation, and the irrigated water was the Yellow River water, with a mineralization degree of nearly 0.67 g/L, which was taken out by a pump, and the irrigated amount of water was measured and controlled by a water meter.

### 2.3. Data Collection and Measurement Methods

#### 2.3.1. Meteorological Data

We set up a micrometeorological station (HOBO-U30, Bourne, MA, USA) in the experimental area to automatically record meteorological data, including temperature, precipitation, wind speed, air humidity, solar radiation, sunshine hours, and solar radiation. We calculated crop evapotranspiration ($ET_0$) using the Penman–Monteith formula [31]. The rainfall during the growth period of oilseed sunflowers in 2020 and 2021 was 137.9 mm and 91.2 mm, respectively. We calculated the potential evapotranspiration of crop as $ET_p = K_c \times ET_0$, where Kc is the oilseed sunflower crop coefficient. According to the Food and Agriculture Organization of the United Nations (FAO56) [31], the recommended values in the early, middle, and late growth stages are 0.2, 1.1, and 0.45, respectively. In the

HYDRUS-2D model, the potential evapotranspiration ($ET_p$) was divided into potential evaporation ($E_p$) and potential transpiration ($T_p$) (Figure 2), and the calculation equations [32] are as follows:

$$ET_p = T_p + E_p \tag{1}$$

$$E_p = e^{-k \cdot LAI} ET_p \tag{2}$$

$$T_p = \left(1 - e^{-k \cdot LAI}\right) ET_p \tag{3}$$

where $k$ is the extinction coefficient, and $LAI$ is the leaf area index.

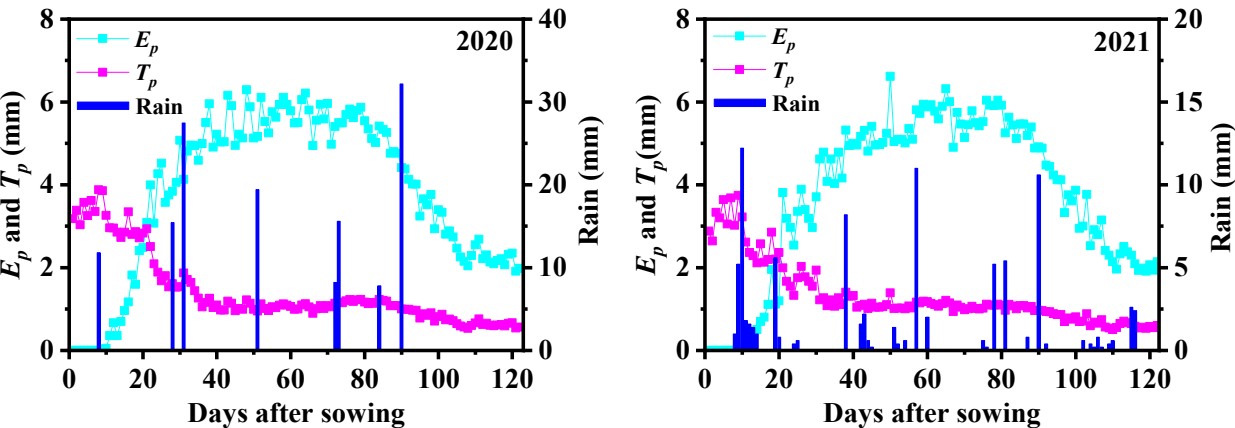

**Figure 2.** Rain, potential transpiration ($T_p$) and potential evaporation ($E_p$) in the 2020 and 2021 oilseed sunflower growing seasons.

### 2.3.2. Soil Water Content

A soil drill (Beijing New Landmark Soil Equipment Co., Beijing, China) was used to collect soil samples, with a total of six layers (0–10 cm, 10–20 cm, 20–40 cm, 40–60 cm, 60–80 cm, and 80–100 cm). Determination of soil moisture content was performed using the drying method. The sampling interval was 5–7 d.

### 2.3.3. Soil NO₃-N Concentration

The sampling location and sampling interval of soil nitrate nitrogen concentration were the same as those of SWC. We used the semi-micro Kjeldahl method to determine the concentration of nitrate nitrogen in the soil; the soil samples were mixed and shaken with $2 \text{ mol L}^{-1}$ KCl solution (soil-to-liquid ratio of 1:5) and analyzed using a UV spectrophotometer (Beijing General Instrument Co., Ltd., TU-1901, General Instrument, Beijing, China). At the same time, the total nitrogen absorption of sunflower stems, leaves, and seeds were measured. These samples were first killed at a temperature of 105 °C and then stored in an oven at 75 °C to reach a constant weight. After crushing and sieving, 0.2 g of the sample was weighed with paper, digested with 5 mL of concentrated $H_2SO_4$, and measured using a flow analyzer (Brown Ruby Inc., AA3, SEAL, Germany). We determined the $NO_3$-N uptake using the semimicro Kjeldahl method.

Before sowing oilseed sunflower, we measured the vertical water flux and soil solution using a self-made lysimeter and a PVC pipe with an opening at the bottom (installed at 100 cm depth). The sampling interval was 7–15 d. We multiplied the $NO_3$-N concentration in the collected soil $NO_3$-N by the corresponding vertical water flux to determine the cumulative $NO_3$-N leaching for a specific time interval.

### 2.3.4. Plant Leaf Area, Height, and Yield of Oilseed Sunflower

The plant heights of oilseed sunflowers were determined using a tape measure (with an accuracy of 0.1 cm). The leaf areas of the oilseed sunflowers were determined using a

leaf area measuring instrument (Li-3000, LI-COR, Lincoln, NE, USA). The measurement interval was 7–10 d. The leaf area index was calculated using the FAO method [31]. The roots of oilseed sunflower were collected at the seedling, budding, flowering, and mature stage. After cleaning, the roots were scanned using WinRHIZO LA2400 software.

When the oilseed sunflower was mature, we selected 20 standard sample plants from non-side rows in each plot and harvested them separately for seed testing and yield testing.

### 2.4. HYDRUS Model Modeling and Input Parameters

#### 2.4.1. Basic Principles of the Model

Simulation of soil water flow and solute transport in oil sunflower fields with concealed drainage using the HYDRAS-2D model [33]. We solved the model using the finite element method [34]. The equation is as follows:

$$\frac{\partial \theta}{\partial t} = \frac{\partial}{\partial x}\left[K(h)\frac{\partial h}{\partial x}\right] + \frac{\partial}{\partial z}\left[K(h)\frac{\partial h}{\partial z} + K(h)\right] + S(h) \tag{4}$$

where $h$ is the pressure head, cm; $\theta$ is the soil volumetric water content, $cm^3/cm^3$; $t$ is the simulation time, d; $K(h)$ is the hydraulic conductivity, cm/d; $S$ is the root water uptake, $d^{-1}$; and $x$ and $z$ are the horizontal and vertical coordinates, cm.

The soil hydraulic function adopts the van Genuchten (1980) model formula, as follows:

$$\theta = \theta_r + \frac{\theta_s - \theta_r}{\left(1 + (\alpha|h|)^n\right)^m}\left(m = 1 - \frac{1}{n}\right) \tag{5}$$

$$K(h) = K_s S_e{}^l\left[1 - \left(1 - S_e^{\frac{1}{m}}\right)^m\right]^2 \tag{6}$$

$$S_e = \frac{\theta - \theta_r}{\theta_s - \theta_r} \tag{7}$$

where $\theta_s$ is the saturated hydraulic conductivity, $cm^3/cm^3$; $\theta_r$ is the residual soil water content; $S_e$ is the relative saturation; $n$, $m$, and $\alpha$ are the shape parameters; $K_s$ is the saturated hydraulic conductivity, cm/d; and l is the pore correlation parameter, $l = 0.5$ [35].

The solute transport equation of $NH_4$-N and $NO_3$-N considers the convection–dispersion effect in the liquid phase. In this study, the partial differential equation for controlling the non equilibrium transport of solutes in the continuous first order decay chain during transient water flow in a variable saturated rigid porous medium is simplified as follows:

for $NH_4$-N:

$$\frac{\partial \theta c_1}{\partial t} + \rho\frac{\partial s_1}{\partial t} =$$
$$\frac{\partial}{\partial x}\left(\theta D_{xx}\frac{\partial c_1}{\partial x}\right) + \frac{\partial}{\partial x}\left(\theta D_{xz}\frac{\partial c_1}{\partial z}\right) + \frac{\partial}{\partial z}\left(\theta D_{zx}\frac{\partial c_1}{\partial x}\right) + \frac{\partial}{\partial z}\left(\theta D_{zz}\frac{\partial c_1}{\partial z}\right) - \left(\frac{\partial q_x c_1}{\partial x} + \frac{\partial q_z c_1}{\partial z}\right) \tag{8}$$
$$-\mu_l \theta c_1 - \mu_s \rho s_1 - S_c 1$$

for $NO_3$-N:

$$\frac{\partial \theta c_2}{\partial t} = \frac{\partial}{\partial x}\left(\theta D_{xx}\frac{\partial c_2}{\partial x}\right) + \frac{\partial}{\partial x}\left(\theta D_{xz}\frac{\partial c_2}{\partial z}\right) + \frac{\partial}{\partial z}\left(\theta D_{zx}\frac{\partial c_2}{\partial x}\right) + \frac{\partial}{\partial z}\left(\theta D_{zz}\frac{\partial c_2}{\partial z}\right) - \left(\frac{\partial q_x c_2}{\partial x} + \frac{\partial q_z c_2}{\partial z}\right)$$
$$+ \theta\mu c_1 - S_c 2 \tag{9}$$

where $\rho$ is the soil bulk density ($g \cdot cm^{-3}$); $\theta$ is the soil water content ($cm^3 \cdot cm^{-3}$); $s_1$ is the mass concentration of $NH_4$-N in the soil ($mg \cdot g^{-1}$); $c$ is the solute concentration of $NH_4$-N and $NO_3$-N in the liquid phase; $D_{xx}$, $D_{xz}$, $D_{zx}$, and $D_{zz}$ are the effectively dispersed components of the coefficient tensor ($cm^2 \cdot d^{-1}$); $\mu_l$ and $\mu_s$ are the first-order reaction rate constants, representing the nitrification process in the liquid and solid phases, respectively

($d^{-1}$); $q_x$ and $q_z$ are the components of the volumetric flux density ($cm^2 \cdot d^{-1}$); and $S$ is the sink term ($d^{-1}$). Equations (10) and (11) usually include the solute flux because of dispersion, solute flux caused by water convection, and nitrogen uptake by roots. $Sc$ ($mg \cdot cm^{-3} \cdot d^{-1}$) is related to the root water uptake $S$:

$$S_c1 = S(h)c_1 \tag{10}$$

$$S_c2 = S(h)c_2 \tag{11}$$

where $c_1$ and $c_2$ represent the concentrations of $NH_4$-N and $NO_3$-N absorbed by roots ($mg \cdot cm^{-3}$), respectively.

### 2.4.2. Root Water Uptake

The root water absorption term adopts the Feddes model [36], as follows:

$$S(h) = \alpha(h) \cdot \beta(x,z) \cdot T_p \cdot L_t \tag{12}$$

where $\beta(x, z)$ is the distribution parameter of root water uptake, $cm^{-2}$; $\alpha(h)$ is the root water pressure function, ($0 < \alpha < 1$); $Tp$ is the potential evaporation rate, cm/d; and Lt is the soil surface width related to crop transpiration process, cm. The water pressure function is as follows [36]:

$$\alpha(h) = \begin{cases} \frac{h_1 - h}{h_1 - h_2} & h_2 < h \leq h_1 \\ 1 & h_3 \leq h \leq h_2 \\ \frac{h - h_4}{h_3 - h_4} & h_4 \leq h < h_3 \end{cases} \tag{13}$$

where $h_1$ is the pressure head at the anaerobic point of root water uptake, cm, taken as $-15$ cm; $h_2$ is the optimum pressure head for root water uptake, $-350$ cm; $h_3$ is the pressure head at the end of root water uptake, $-700$ cm; and $h_4$ is the pressure head at the wilting point of root water uptake, $-8000$ cm. The crop-specific values of oilseed sunflower were taken from the HYDRUS-2D database [37].

### 2.4.3. Initial and Boundary Conditions

The modeling area was a rectangle with a width of 120 cm and a deepness of 170 cm, and the maximum measured depth of groundwater was 167 cm. The top boundary of the modeling area was divided into the covered membrane zone and the uncovered membrane zone, where the covered membrane boundary was set as the no-flux boundary, and the uncovered membrane boundary was set as the atmospheric boundary. The left and right boundaries of the modeling area were set as impermeable boundaries, the down boundary was set as a variable headed boundary, the groundwater head was assigned day by day, and the boundary of the concealed pipe was set as a seepage boundary, as shown in Figure 1. Non-uniform finite element grids produced by the HYDRUS-2D model were used to discretize the simulation area in time and space. Soil water content and $NH_4$-N and $NO_3$-N contents measured before the start of each year's experiment were used as initial conditions for the simulation. The water content at the top and bottom of the modeling area was 0.29 $cm^3 \cdot cm^{-3}$ and 0.38 $cm^3 \cdot cm^{-3}$. The initial values of $NH_4$-N and $NO_3$-N reflect only the initial application of the basal fertilizer because of the relatively thorough $NO_3$-N drenching due to spring irrigation prior to planting. Initial water content and solute concentration levels were assumed to be uniformly distributed. For solute transport the lower boundary and the subsurface pipe boundary were set as Type III boundaries, and the left and right boundaries were set as no flux boundaries. For CD, the modeling results of the previous stage were assigned as the initial input conditions for the next stage one by one by node, and the variable time step profiling method was used to adjust the time-step according to the number of convergence iterations.

### 2.4.4. Model Parameters

The soil hydraulic parameters ($\theta_s$, $\theta_r$, $n$, $m$, $\alpha$, and $K_s$) were determined based on the dry bulk density of the soil and the content of soil sand, silt, and clay particles, which were predicted using the Rosetta function (Table 2). Further calibration of soil hydraulic parameters by comparing simulated and observed values of water and NO$_3$-N content through observation. The longitudinal dispersions of soil at 0–20 cm, 20–40 cm, 40–60 cm, 60–80 cm, and 80–100 cm depths were set to 21 cm, 18 cm, 16 cm, 10 cm, and 8 cm, respectively. The lateral dispersion was set to one-tenth of the longitudinal dispersion [38]. The molecular diffusivities of NH$_4$-N and NO$_3$-N were set to 1.316 cm$^2\cdot$d$^{-1}$ and 1.422 cm$^2\cdot$d$^{-1}$, respectively. Finally, by comparing simulated and observed values of nitrogen uptake by crops, we obtained the maximum allowable concentration of nutrient uptake by crop roots, csmax.

**Table 2.** The 0–100 cm soil parameters in the model. $\theta_r$ is the residual soil water content, $\theta_s$ is the saturated soil water content, $\alpha$ and $n$ ($-$) are shape parameters, $K_s$ is the saturated hydraulic conductivity, and $l$ is the tortuosity parameter in the conductivity function ($-$).

| Soils Layer (cm) | $\theta_r$ (cm$^3\cdot$cm$^{-3}$) | $\theta_s$ (cm$^3\cdot$cm$^{-3}$) | $\alpha$ (cm$^{-1}$) | $n$ (-) | $K_s$ (cm$\cdot$d$^{-1}$) | $l$ ($-$) |
|---|---|---|---|---|---|---|
| 0–20 | 0.0815 | 0.4845 | 0.006 | 1.6429 | 3.40 | 0.5 |
| 20–40 | 0.0747 | 0.5072 | 0.0058 | 1.6493 | 3.28 | 0.5 |
| 40–60 | 0.0712 | 0.4922 | 0.0063 | 1.6327 | 1.07 | 0.5 |
| 60–80 | 0.0624 | 0.4716 | 0.0066 | 1.632 | 1.79 | 0.5 |
| 80–100 | 0.0748 | 0.4675 | 0.0065 | 1.6243 | 4.47 | 0.5 |
| 100–120 | 0.0741 | 0.4871 | 0.0075 | 1.6165 | 3.77 | 0.5 |

Solute transport in the HYDRUS (2D) model is a relatively complex process, in which nitrogen transport may include processes, such as nitrification, denitrification, ammonia volatilization, solidification, and mineralization. Because denitrification occurs under saturated conditions, we ignored the denitrification process is ignored [30,39]. In addition, similarly to other studies, considering the NO$_3$-N transport several days after irrigation and fertilization [22], we also ignored the processes of solidification and mineralization. At the same time, ammonia volatilization was also ignored because of the simultaneous application of fertilizers with irrigation water [40]. Conversely, the model was used to simulate NO$_3$-N at the time of urea application. In this study, we assumed that urea undergoes instantaneous nitrification to NO$_3$-N, which was a reasonable assumption because nitrification is faster than other effects and requires several days [22]. In this study, it was also hypothesized that NO$_3$-N was not adsorbed and was exclusively contained in the dissolved phase, whereas NH$_4$-N was adsorbed and was present in both the solid and dissolved phases. The partition coefficients (K$_d$) for NO$_3$-N and NH$_4$-N were, respectively, set to 0 and 2.9 cm$^3$ g$^{-1}$, and the first-order rate of solute (nitration) constants for the liquid and solid phases were set to 0.03 and 0.13 d$^{-1}$.

### 2.4.5. Evaluation of Model Properties

Parameters were calibrated using the 2020 soil water content and nitrogen content and validated with 2021 data. Model rate-setting and testing were evaluated using mean relative error (*MRE*), root mean square error (*RMSE*), and coefficient of determination (*R$^2$*) for testing.

$$MRE = \frac{1}{n}\sum_{i=1}^{n}\frac{S_i - M_i}{M_i} \times 100\% \tag{14}$$

$$RMSE = \sqrt{\frac{1}{n}\sum_{i=1}^{n}(S_i - M_i)^2} \tag{15}$$

$$R^2 = 1 - \frac{\sum_{i=1}^{n}(S_i - M_i)^2}{\sum_{i=1}^{n}(M_i - \overline{M})^2} \tag{16}$$

where $S_i$ is the simulated value; $M_i$ is the actual measured value; $i$ is the $i$-th observation point; $n$ is the total number of observation points; and $\overline{M}$ is the average actual measured value.

## 3. Results

### 3.1. Calibration, Validation, and Performance Evaluation of the Model

The parameters of the HYDRAUS-2D model were manually calibrated using measurement data from 2020 and validated using corresponding data from 2021 (Table 3). The HYDRUS-2D model effectively simulated the dynamic changes of the soil water content and NO$_3$-N in oilseed sunflower farmland under different drainage conditions. The simulation results of water content and NO$_3$-N content in the 0–100 cm soil layer were in good agreement with the measured values. Fine sands (dune sands) were added to the soil in this experiment to increase soil permeability and accelerate water and solute transport in the soil profile. During the calibration process, the MREs of the simulated and measured soil water content and NO$_3$-N content were 8.45–11.22% and 10.08–13.31%, respectively; the *RMSE*s were 0.03–0.06 cm$^3 \cdot$cm$^{-3}$ and 10.25–11.52 mg$\cdot$kg$^{-1}$, respectively; and the coefficients of determination ($R^2$) were 0.80–0.92 and 0.77–0.89, respectively. During the validation process, the *MRE*s of the simulated and measured values of soil water content and NO$_3$-N content were 8.44–12.78% and 10.25–14.25%, respectively; the *RMSE*s were 0.02–0.07 cm$^3 \cdot$cm$^{-3}$ and 10.14–11.87 mg$\cdot$kg$^{-1}$, respectively; and the coefficients of determination ($R^2$) were 0.78–0.85 and 0.74–0.86, respectively. The results showed that the HYDRUS-2D model sufficiently captured the dynamic changes of soil water content and NO$_3$-N content in time and space. Therefore, the HYDRAS-2D model can effectively simulate the soil water content and NO$_3$-N transport under different controlled drainage conditions.

**Table 3.** Evaluation of simulation accuracy for soil water content (SWC) and NO$_3$-N nitrogen content (NC) in 0–100 cm soil in the 2020 and 2021.

| Treatment | Parameter | Error | 2020 (Calibration) | | | | 2021 (Validation) | | | |
|---|---|---|---|---|---|---|---|---|---|---|
| | | | 0–20 cm | 20–40 cm | 40–60 cm | 60–100 cm | 0–20 cm | 20–40 cm | 40–60 cm | 60–100 cm |
| FD | SWC | MRE (%) | 9.84 | 8.45 | 8.75 | 9.15 | 9.15 | 8.45 | 9.12 | 8.44 |
| | | RMSE (cm$^3 \cdot$cm$^{-3}$) | 0.05 | 0.05 | 0.04 | 0.03 | 0.04 | 0.06 | 0.03 | 0.05 |
| | | $R^2$ | 0.92 | 0.90 | 0.88 | 0.87 | 0.83 | 0.81 | 0.84 | 0.85 |
| | NC | MRE (%) | 12.24 | 11.24 | 10.08 | 11.72 | 10.25 | 12.19 | 13.38 | 12.75 |
| | | RMSE (mg$\cdot$kg$^{-1}$) | 11.52 | 10.02 | 10.52 | 11.11 | 10.52 | 10.65 | 10.14 | 11.11 |
| | | $R^2$ | 0.80 | 0.85 | 0.83 | 0.89 | 0.86 | 0.76 | 0.82 | 0.85 |
| CWT1 | SWC | MRE (%) | 10.52 | 10.58 | 10.76 | 11.22 | 12.78 | 12.12 | 10.19 | 9.83 |
| | | RMSE (cm$^3 \cdot$cm$^{-3}$) | 0.03 | 0.05 | 0.04 | 0.06 | 0.02 | 0.07 | 0.04 | 0.05 |
| | | $R^2$ | 0.86 | 0.84 | 0.89 | 0.82 | 0.80 | 0.82 | 0.85 | 0.81 |
| | NC | MRE (%) | 12.25 | 13.14 | 11.85 | 13.31 | 10.85 | 11.12 | 12.35 | 13.33 |
| | | RMSE (mg$\cdot$kg$^{-1}$) | 11.22 | 11.42 | 11.25 | 10.78 | 10.29 | 10.83 | 11.11 | 11.25 |
| | | $R^2$ | 0.79 | 0.81 | 0.82 | 0.79 | 0.74 | 0.75 | 0.82 | 0.77 |
| CWT2 | SWC | MRE (%) | 10.44 | 9.63 | 9.82 | 9.85 | 10.81 | 10.93 | 11.82 | 11.53 |
| | | RMSE (cm$^3 \cdot$cm$^{-3}$) | 0.04 | 0.05 | 0.04 | 0.06 | 0.07 | 0.04 | 0.04 | 0.02 |
| | | $R^2$ | 0.83 | 0.80 | 0.81 | 0.85 | 0.80 | 0.82 | 0.78 | 0.81 |
| | NC | MRE (%) | 12.24 | 11.58 | 12.54 | 13.14 | 14.25 | 12.48 | 12.11 | 13.29 |
| | | RMSE (mg$\cdot$kg$^{-1}$) | 11.06 | 10.83 | 10.42 | 10.58 | 10.65 | 11.82 | 11.64 | 11.87 |
| | | $R^2$ | 0.77 | 0.80 | 0.82 | 0.85 | 0.84 | 0.80 | 0.80 | 0.78 |

### 3.2. Effect of CD on the Dynamic Changes of Soil Nitrate

Figure 3 shows an analysis of the soil $NO_3$-N content at the center of the two subsurface drains in the farmland plot. The analysis indicated that after the application of nitrogen fertilizer, the soil $NO_3$-N content increased immediately, which significantly increased the $NO_3$-N content, and resulted in two peaks during the application of basal fertilizer and topdressing during the budding stage. From the sowing to budding stage (DAS (Days after sowing) 0–47 in 2020, DAS 0–52 in 2021) before irrigation, the rainfall in 2020 and the relatively low soil permeability coefficient in the experimental area affected the $NO_3$-N content in the surface soil. We did not observe any significant difference in the $NO_3$-N content in the 0–40 cm soil layer among the treatments. The $NO_3$-N contents of the CWT1, CWT2, and FD treatments were 4.82, 4.26, and 3.81 mg·kg$^{-1}$ in the 0–20 cm soil layer and 6.52, 5.73, and 6.26 mg·kg$^{-1}$ for the 20–40 cm soil layer, respectively. There were nearly no differences among the treatments for other time points or soil depths.

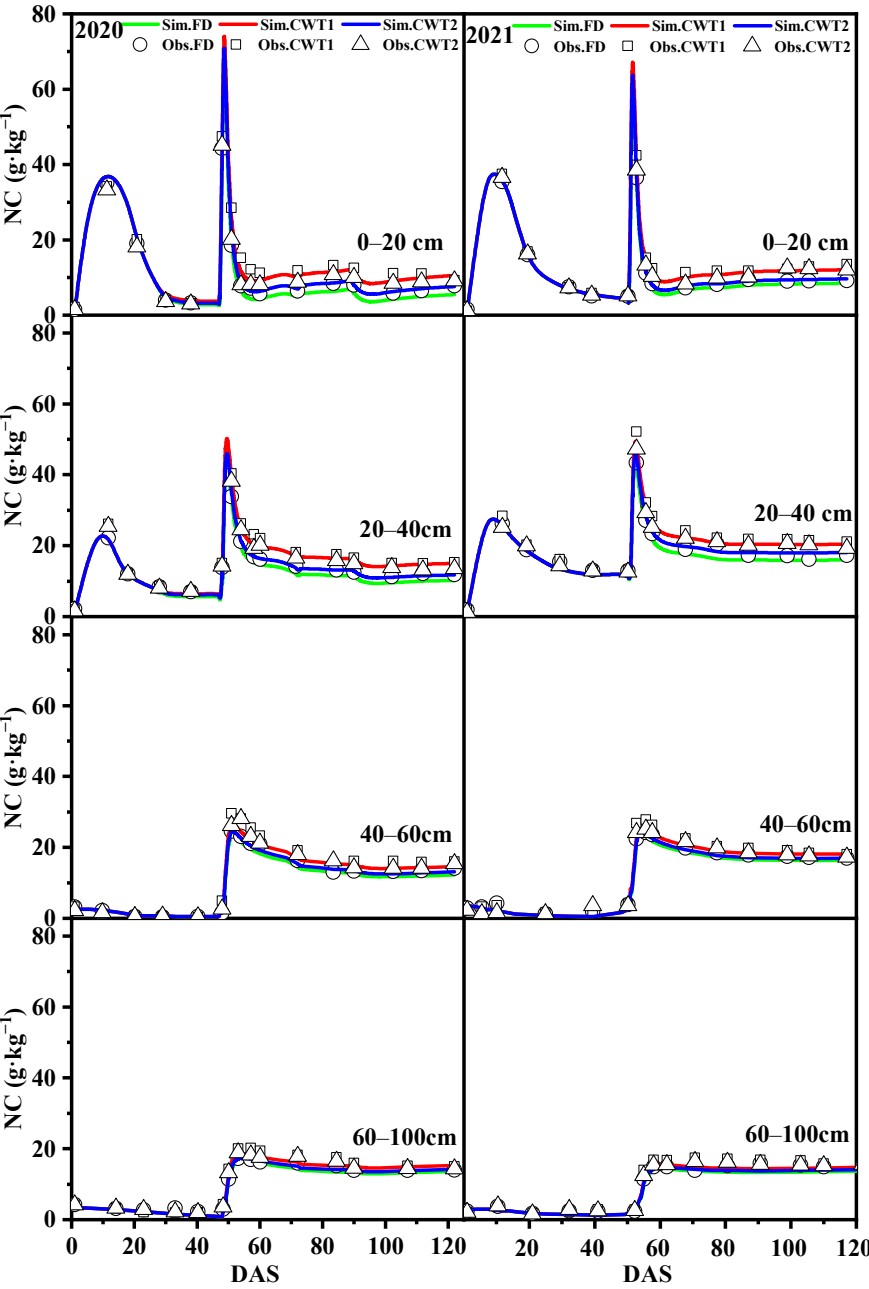

**Figure 3.** Simulated and observed the dynamic changes of $NO_3$-N content (NC) in the 0–100 cm in 2020 (**left**) and 2021 (**right**).

From irrigation during the growth stage to harvesting (DAS 48–122 in 2020, DAS 53–122 in 2021), because of the effect of CD and the relatively large amount of irrigation during the growth stage, the soil $NO_3$-N content exhibited significant differences between treatments. In 2020, from irrigation during the growth stage to harvesting, the soil $NO_3$-N contents in the 0–20 cm, 20–40 cm, 40–60 cm, and 60–100 cm soil layers of the CWT1 treatment were 28.83%, 25.04%, 16.10%, and 13.37% higher than those of the FD treatment, respectively. Those of the CWT2 treatment, however, were 12.82%, 8.98%, 5.74%, and 4.77% higher than those of the FD treatment, respectively. In 2021, from irrigation during the growth stage to harvesting, the $NO_3$-N content in the 0–100 cm soil layer was higher for CWT1 (13.66%) and CWT2 (5.49%) treatments compared to that of FD. From irrigation during the growth period to harvesting sunflowers, the average soil $NO_3$-N contents of the CWT1, CWT2, and FD treatments were 22.10 mg·kg$^{-1}$, 20.51 mg·kg$^{-1}$, and 19.45 mg·kg$^{-1}$, respectively.

There is a significant difference in soil $NO_3$-N content after irrigation in the budding stage of oilseed sunflower fields under controlled drainage, with the largest difference seen for the 0–40 cm soil layer. After irrigation in the growth stage in 2020 and 2021, the average soil $NO_3$-N contents in the 0–40 cm of the FD, CWT1, and CWT2 treatments were 18.09 mg·kg$^{-1}$, 21.15 mg·kg$^{-1}$, and 19.29 mg·kg$^{-1}$, respectively. CWT1 and CWT2 treatments were 16.94% and 6.66% above the FD treatment, respectively. CD ensured sufficient nutrient uptake and utilization of oilseed sunflower, which provided a more suitable growth environment for oilseed sunflower and improved soil nutrient availability.

*3.3. Effects of Drainage Methods on Soil Nitrate Distribution*

The changes in soil $NO_3$-N content after irrigation and rainfall in 2020 and 2021 were similar. Hence, only the 2021 results are discussed here, as the second year of the CD experiment was more representative. The FD, CWT1, and CWT2 treatments greatly affected the vertical and horizontal distributions of $NO_3$-N content in the soil profile after irrigation at the budding stage. Figure 4 presents the distributions of $NO_3$-N in 2021 at different distances from the subsurface drain and different soil depths in DAS52 (i.e., one day before nitrogen application in the growth stage; Figure 4a–c), DAS54 (i.e., one day after nitrogen application in the growth stage; Figure 4d–f), DAS58 (five days after irrigation and nitrogen application in the growth stage; Figure 4g–i), and DAS78 (the late growth stage of oilseed sunflower; Figure 4j–l). Because of a small amount of precipitation before irrigation during the growth stage, the distributions of $NO_3$-N content in the soils of different treatments were almost the same, and any of these differences were small. The $NO_3$-N content in the 0–40 cm soil layer was relatively high at 3.32–11.96 mg·kg$^{-1}$, whereas that in the 40–100 cm soil layer ranged from 1.3 to 2.66 mg·kg$^{-1}$. The $NO_3$-N content in the 0–20 cm soil layer in the growth stage (DAS54) increased immediately one day after the irrigated nitrogen application, reaching a peak of 50.28–68.03 mg·kg$^{-1}$. The soil $NO_3$-N content of the CD treatment was significantly higher than that of the FD treatment. The CWT1 and CWT2 treatments were 14.01% and 5.75% higher than the FD treatments, respectively. Five days after irrigation and nitrogen application (DAS58), the soil $NO_3$-N contents of different treatments were significantly different. The soil $NO_3$-N contents in the 0–20 cm, 20–40 cm, 40–60 cm, and 60–100 cm soil layers of the CWT1 treatment were 15.66%, 19.23%, 15.51%, and 9.03% higher than those of the FD treatment, respectively. The CWT2 treatments, however, were 1.94%, 8.55%, 2.21%, and 0.87% higher than those of the FD treatment, respectively. Significant spatial differences were observed; that is, the soil $NO_3$-N content increased with the distance from the subsurface drain. For the FD, CWT1, and CWT2 treatments, the $NO_3$-N contents in the center point were 52.35%, 25.76%, and 43.48% higher than those in the 0–100 cm soil layer, respectively. In the late growth stage (DAS78), the difference in soil $NO_3$-N content between treatments became smaller, and the soil $NO_3$-N content was lower (4.58–21.03 mg·kg$^{-1}$). The difference in $NO_3$-N content of each treatment at different distances from the subsurface drain also became smaller. This meant that the CD treatment created a good environment for crop growth after irrigation

and nitrogen application during the crop growth stage, which promoted crop nutrient uptake and increased crop yield.

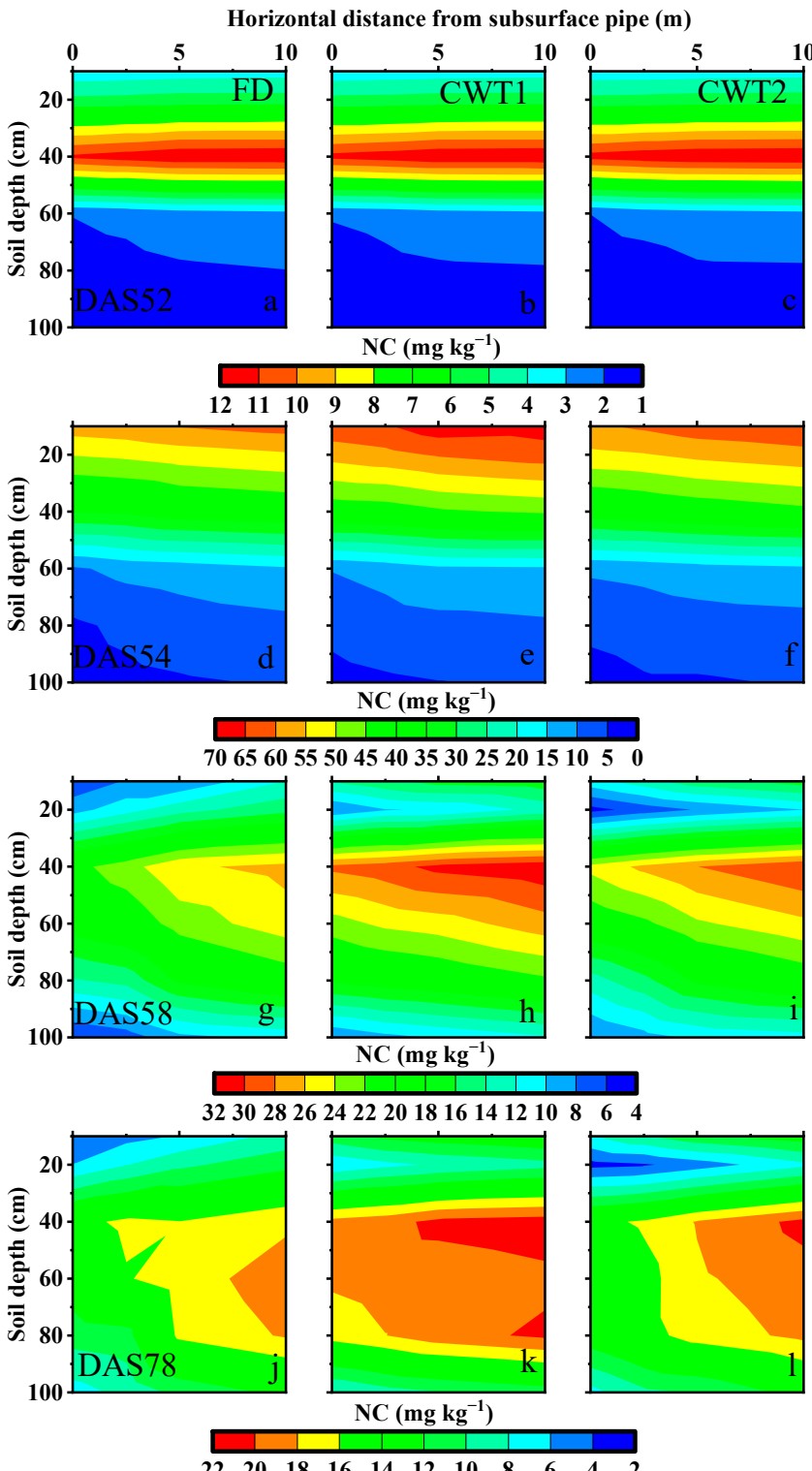

**Figure 4.** Two-dimensional distributions of nitrate nitrogen content (NC) in a field under free drainage (FD, on the left, (**a**,**d**,**g**,**j**)), controlled drainage depth of 40 cm (CWT1, in the middle, (**b**,**e**,**h**,**k**)), controlled drainage depth of 70 cm (CWT2, on the right, (**c**,**f**,**i**,**l**)), on DAS 52 (1 day before irrigation and nitrogen application, (**a–c**)), DAS 54 (1 day after irrigation and nitrogen application (**d–f**)), DAS 58 (5 days after irrigation and nitrogen application, (**g–i**)), and DAS78 (late growth stage, (**j–l**)).

### 3.4. Effects of Drainage Methods on Nitrogen Uptake, Leaching, and Loss in Farmland

In the early stage of crop growth (DAS 0–30 in 2020, DAS 0–35 in 2021), the simulated and measured crop nitrogen uptake values were relatively low. In the middle stage of crop growth (DAS 30–92 in 2020, DAS 35–96 in 2021), they increased rapidly. They declined again in the late growth stage (after DAS 92 in 2020, after DAS 96 in 2021) (Figure 5a,b). From 2020 to 2021, the average daily nitrogen uptake intensities of the FD, CWT1, and CWT2 treatments were 1.41 $kg \cdot ha^{-1} \cdot d^{-1}$, 1.51 $kg \cdot ha^{-1} \cdot d^{-1}$, and 1.46 $kg \cdot ha^{-1} \cdot d^{-1}$, respectively. The average daily nitrogen uptake intensities in the early stage of crop growth were 0.83 $kg \cdot ha^{-1} \cdot d^{-1}$, 0.85 $kg \cdot ha^{-1} \cdot d^{-1}$, and 0.84 $kg \cdot ha^{-1} \cdot d^{-1}$, respectively. In the middle stage of crop growth, the average daily nitrogen uptake intensities were 2.38 $kg \cdot ha^{-1} \cdot d^{-1}$, 2.55 $kg \cdot ha^{-1} \cdot d^{-1}$, and 2.48 $kg \cdot ha^{-1} \cdot d^{-1}$, respectively. In the late crop growth stage, the average daily nitrogen uptake intensities were 0.42 $kg \cdot ha^{-1} \cdot d^{-1}$, 0.45 $kg \cdot ha^{-1} \cdot d^{-1}$, and 0.41 $kg \cdot ha^{-1} \cdot d^{-1}$, respectively. There was little difference in nitrogen uptake among the treatments from sowing to the growth stage. We did observe, however, a significant difference in nitrogen uptake from the growth stage to the harvest of oilseed sunflower among the treatments. The two-year cumulative nitrogen uptake of each treatment was in the order of CWT1 > CWT2 > FD. The cumulative nitrogen uptake of the CD treatment was relatively high, which eventually led to an increase in crop yield (Table 4). The cumulative nitrogen uptake of the CWT1 treatment was 6.65% above that of the FD treatment, and the CWT1 treatment was 3.38% higher than that of the CWT2 treatment.

**Table 4.** Components of the N balance, corn yield, and the N use efficiency (NUE).

| Year | Treatments | Components of N Balance ($kg \cdot ha^{-1}$) | | | | | | | Yield ($kg \cdot ha^{-1}$) | NUE ($kg \cdot ha^{-1}$) |
|------|-----------|---------|---------|--------------|------------|----------|---------|----------|-------|------|
| | | Initial | Applied | Nitrification | Crop Uptake | Leaching | Drained | Residual | | |
| 2020 | FD | 26.43 | 171.8 | 16.02 | 169.37 | 34.3 | 0.58 | 10 | 3562.10 | 21.03 |
| | CWT1 | 26.43 | 171.8 | 20.2 | 181.55 | 25.94 | 0.24 | 10.7 | 3836.11 | 21.13 |
| | CWT2 | 26.43 | 171.8 | 18.28 | 175.43 | 30.39 | 0.29 | 10.4 | 3670.33 | 20.93 |
| 2021 | FD | 30.24 | 171.8 | 18.33 | 175.49 | 33.45 | 0.6 | 10.83 | 3621.57 | 20.64 |
| | CWT1 | 30.24 | 171.8 | 23.32 | 186.25 | 27.26 | 0.33 | 11.52 | 3952.14 | 21.23 |
| | CWT2 | 30.24 | 171.8 | 20.26 | 180.36 | 30.49 | 0.42 | 11.03 | 3768.28 | 20.88 |

Three $NO_3$-N leaching events occurred because of two heavy rainfalls and irrigation during the whole growth stage in 2020. In 2021, only one $NO_3$-N leaching event happened because of irrigation in the growth period. As a result of the high outlet of the subsurface drainage, the groundwater decline was stable, resulting in a small amount of $NO_3$-N leaching. The average amount of $NO_3$-N leached in two years was higher for the CWT1 treatment (27.35%) compared to the CWT2 (11.28%) treatment (Figure 5c,d). CD reduced the amount of $NO_3$-N leaching after heavy rainfall and irrigation and increased the crop nitrogen uptake. FD leached a large amount of $NO_3$-N from the crop root zone, which adversely affected the soil and groundwater environment.

Similar to $NO_3$-N leaching, the FD treatment in 2020 produced three drainage events, whereas the CWT1 and CWT2 treatments each produced only one drainage event after irrigation in the crop growth stage because of the shallow subsurface drain. Among these treatments, the FD treatment caused high $NO_3$-N loss because of drainage. The two-year $NO_3$-N loss of the FD treatment was 107.02% and 66.20% higher than that of the CWT1 and CWT2 treatments, respectively (Figure 5e,f). CD effectively reduced the loss of nitrogen in the soil after nitrogen application during the growth stage and reduced the concentration of nutrients in the drainage, which holds great significance for the protection of water environment.

In summary, the CD at 40 cm (CWT1) stabilized the groundwater depth, reduced the hydraulic gradient of groundwater runoff, decreased the drainage flow rate, and prolonged the retention time of soil moisture in the farmland. The leaching and loss of $NO_3$-N were reduced, which promoted the crop nitrogen uptake and utilization, improved the nitrogen use efficiency, reduced the waste of nitrogen, and played a positive role in protecting the

soil and water environment. Thus, CWT1 was shown to be a suitable drainage method for the experimental area.

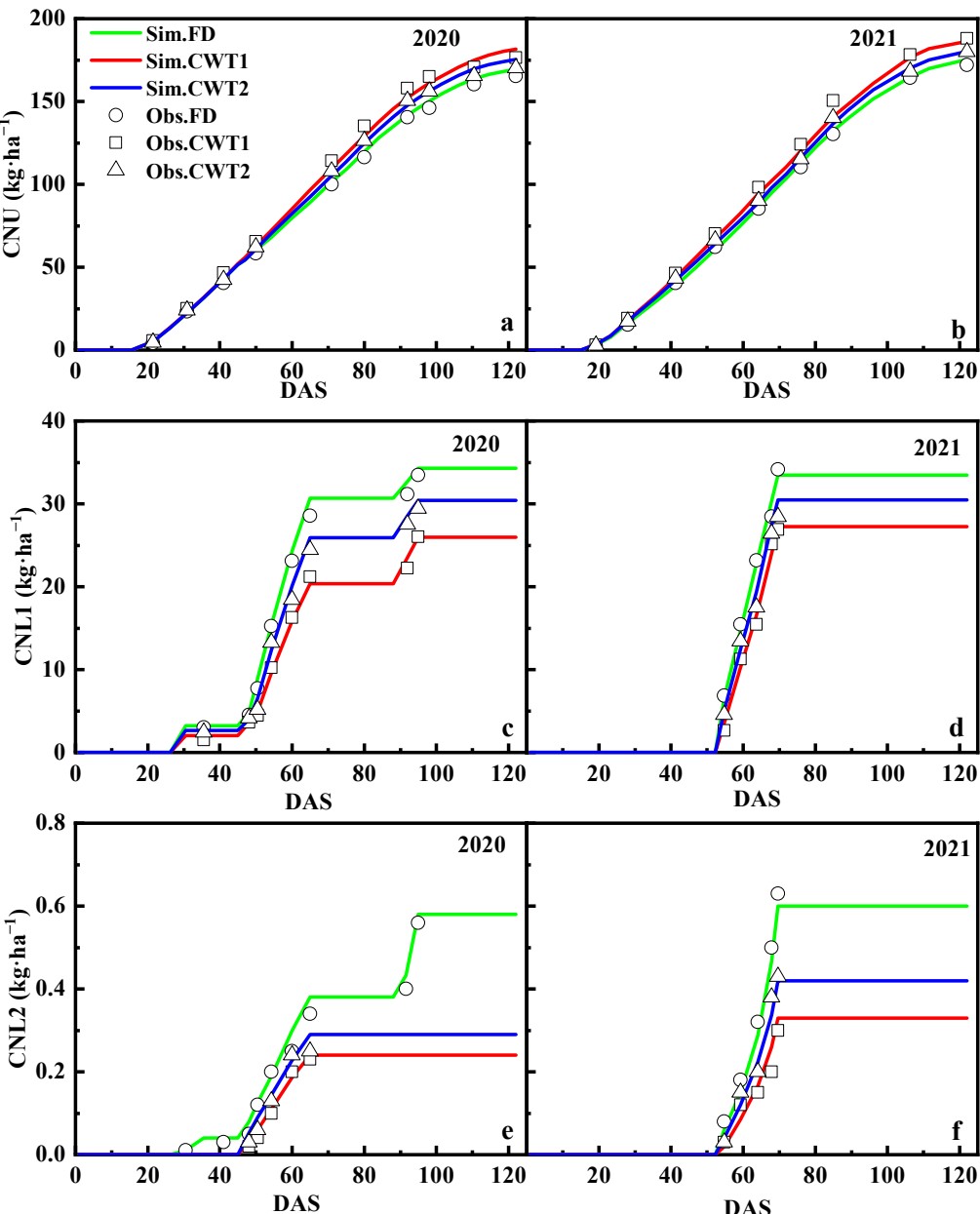

**Figure 5.** Simulated and observed cumulative NO$_3$-N uptake (CNU, top) in 2020 (**a**) and 2021 (**b**), cumulative NO$_3$-N leaching (CNL1, middle) at depths of 100 cm in 2020 (**c**) and2021 (**d**), and cumulative NO$_3$-N loss (CNL2, bottom) from subsurface pipe in 2020 (**e**) and 2021 (**f**).

### 3.5. Effects of Drainage Methods on Nitrogen Balance and Nitrogen Use Efficiency

Table 4 displays the analysis of soil nitrogen balance in the growth stage (from sowing to harvesting) of oilseed sunflower. In 2020 and 2021, the CD treatment enhanced the nitrification intensity. The nitrification rates of the CWT1 and CWT2 treatments increased by 26.70% and 12.20%, respectively, compared with the FD treatment in two years. In 2020 and 2021, the two-year average cumulative nitrogen uptake of the FD, CWT1, and CWT2 treatments were 172.43 kg·ha$^{-1}$, 183.90 kg·ha$^{-1}$, and 177.90 kg·ha$^{-1}$, respectively. CD provided a suitable nutrient environment, which promoted the crop nitrogen uptake and utilization. During the oilseed sunflower growth stage, the two-year average nitrogen leaching amounts of the FD, CWT1, and CWT2 treatments in 2020 and 2021 were

33.88 kg·ha$^{-1}$, 26.60 kg·ha$^{-1}$, and 30.44 kg·ha$^{-1}$, respectively; and the average NO$_3$-N losses through subsurface drainage were 0.59 kg·ha$^{-1}$, 0.29 kg·ha$^{-1}$, and 0.36 kg·ha$^{-1}$, respectively. Apparently, CD reduced the amount of NO$_3$-N leaching and loss during rainfall and irrigation in the crop growth stage.

CD significantly increased the yield of oilseed sunflower. The CWT1 and CWT2 treatments increased the average yields by 4.52% and 3.04%, respectively, relative to the FD treatment ($p < 0.05$). CD also raised the nitrogen use efficiency, with the average nitrogen use efficiencies of the CWT1 and CWT2 treatments increased by 1.66% and 0.34%, respectively, relative to the FD treatment, but the difference was insignificant. After harvesting, the residual NO$_3$-N content in the soil with the CD treatment was relatively high, but the difference was insignificant. The two-year average residual NO$_3$-N contents of the FD, CWT1, and CWT2 treatments were 10.42 kg·ha$^{-1}$, 11.11 kg·ha$^{-1}$, and 10.72 kg·ha$^{-1}$, respectively. These results indicated that CD absorbed and utilized most of the nitrogen in the soil, which had a positive effect on crop growth and yield.

## 4. Discussion

### 4.1. Performance of the HYDRUS-2D Model

Field experiments typically last for a long time and consume a great deal of staffing power and material resources. In addition, field experiment monitoring is limited in terms of space and time. Thus, it is difficult to obtain long-time-series experimental data and challenging to obtain the most complete scientific pattern. Therefore, the study of water and nitrogen cycles in farmland usually involves a combination of field monitoring and theoretical model. The HYDRUS model can be set flexibly according to different boundary conditions and status. It offers a strong advantage in accurately describing the water and solute transport at the specific location of the two-dimensional profile, and increasingly is being adopted in the study of subsurface drainage [41]. Compared with the DRAINMOD, SWAT, and RZWQM2 models, the HYDRUS-2D model is more systematic and comprehensive in simulating soil water and nitrogen transport, crop water and nitrogen uptake, water and nitrogen leaching by irrigation and rainfall, and water and nitrogen discharged through subsurface drain. Thus, this model provides a good basis for further analysis of farmland water and nitrogen balance. Tao et al. (2021) [42] applied the HYDRUS model to simulate nitrogen loss and soil nitrogen content under improved subsurface drainage conditions. They showed that the model performed well in the simulation process.

These studies mainly focused on the soil moisture and solute transport under the subsurface drain boundary and subsurface drainage conditions. In this study, we used the HYDRUS-2D model to conduct a detailed analysis of soil nitrogen balance elements affected by CD under the subsurface drainage conditions. In the calibration period, the average values of MRE, RMSE, and R$^2$ of water content were 10.43%, 0.04 cm$^3$·cm$^{-3}$, and 0.82, respectively, whereas those of NO$_3$-N content were 12.36%, 10.99 mg·kg$^{-1}$, and 0.80, respectively. In the validation period, the average values of MRE, RMSE, and R$^2$ of water content were 10.43%, 0.04 cm$^3$·cm$^{-3}$, and 0.82 cm$^3$·cm$^{-3}$, respectively, whereas those of NO$_3$-N content were 12.36%, 10.99 mg·kg$^{-1}$, and 0.80, respectively. The HYDRAS-2D model can effectively simulate and control the soil water content and NO$_3$-N transport under drainage conditions. This is also because fine sand (dune sands) was added to the soil in this experiment to increase soil permeability and accelerate water and solute transport in the soil profile. Additionally, the soil water content and NO$_3$-N content in the 0–20 cm soil layer were largely affected by irrigation and rainfall. Compared with the 20–100 cm soil layer, the error increased, but all values were within the acceptable range and met the accuracy requirements. Therefore, the HYDRUS-2D model could be used to simulate soil NO$_3$-N transport under CD conditions.

### 4.2. Effects of CD on Soil NO₃-N Content and Crop Uptake

In arid areas, the purpose of controlling drainage during the growth stage is to reduce the loss of water and nutrients and to provide sufficient water and nutrients for crop growth after irrigation and fertilization. Appropriate water and nitrogen conditions not only can improve crop yield and dry matter accumulation, but also can enhance water and fertilizer use efficiencies [43]. Consistent with Wesstrom et al. (2014) [18], this study showed that the soil NO₃-N content with CD was significantly greater than that with FD during the growth stage, and its crop nitrogen uptake was also significantly greater than that with FD. Moreover, the daily nitrogen uptake intensity of nitrogen was also enhanced under CD conditions because the oilseed sunflower was in a state of water and nutrient deficiencies at the late growth stage. CD provided more nutrients and water for crop uptake and utilization, reducing water stress and improving water and fertilizer use efficiencies, ultimately leading to an increase in crop yield.

The main aim of irrigation during the growing period is to provide crops with enough water and nutrients in a timely manner to meet the growing conditions of the crops, thus avoiding yield losses due to lack of nutrients during the growing phase of the crops, which are very insensitive to water resources and nutrients. CD can provide better, more uniform, and more stable surface soil moisture and nutrient conditions, which is particularly significant during a dry summer year by effectively improving deep soil moisture and nutrient conditions. In the late growth stage, the difference in soil NO₃-N content decreased, indicating that the CD treatment enhanced crop nutrient uptake and created a good growth environment for crop growth. During the growth stage, CD of 40 cm (K1) stabilized the soil NO₃-N content difference in horizontal direction.

This study demonstrated that CD can change the NO₃-N content in both vertical and horizontal directions. The NO₃-N content of the FD treatment was quite different from different distances of the subsurface drain, and the CD treatment reduced this difference, so that the crops could grow evenly because of the small difference in the growth environment of the distance from the subsurface drain, reducing the competition for water and fertilizer. CD can retain water and fertilizer while reducing the difference in nutrients between vertical and horizontal directions.

### 4.3. Effect of CD on NO₃-N Leaching and Loss

Previous studies have shown that nitrogen application is an important factor leading to NO₃-N leaching, and irrigation is a necessary condition for NO₃-N leaching. With the increase in nitrogen application and irrigation, the risk of nitrate leaching is greatly increased [44]. This study showed that after irrigation and fertilization during the growth period, CD stabilized the variation of groundwater depth and reduced the hydraulic gradient of runoff. The drainage flow rate declined, the ability of water flow to carry nitrogen was weakened, and the retention time of soil moisture in the farmland was prolonged, which reduced the amount of NO₃-N leaching. For FD, because of the deep buried depth of the subsurface drain, the groundwater decreased rapidly. Therefore, with the decline of groundwater, the soil NO₃-N leached into the deep soil and groundwater. This is one of the reasons why crops are able to absorb more nutrients in the CD treatment.

The response of nitrogen loss to CD depends on the nitrogen concentration and the amount of drainage, and therefore, CD has different effects on nitrogen loss than FD [17]. Most studies have suggested that CD can reduce drainage volume and NO₃-N concentration, resulting in less NO₃-N loss through subsurface drainage [45–48]. The results of our study are consistent with these previous studies. We found that the CWT1 treatment retained more nitrogen in the soil after irrigation and fertilization during the growth stage. Furthermore, the soil in the experimental area was sticky and the water infiltration rate was slow. The nitrogen loss was small and could be maintained in the root layer soil (0–40 cm) for a long time, so that nitrogen could be absorbed and utilized by oilseed sunflower. Increasing the height of the drainage outlet during the growth stage also had a positive effect on water and soil environmental protection. Thus, in arid areas, we

found that CD would be a suitable drainage method for improving water and fertilizer utilization efficiencies and protecting the environment. CD not only reduced the leaching of $NO_3$-N caused by irrigation and fertilization but also reduced the loss of $NO_3$-N. Hence, the content of $NO_3$-N in the soil was higher than that of FD, which provided a suitable growth environment for crops. This environment increased $NO_3$-N uptake, increased crop yield, and improved nitrogen use efficiency.

## 5. Conclusions

Based on the HYDRUS (2D) model, in this study, we evaluated the dynamic changes of soil $NO_3$-N in FD, CD at 40 cm during growth stage (CWT1), and CD at 70 cm during the growth stage (CWT2). We performed the model calibration and validation using the measured data in 2020 and 2021, respectively. The model reasonably simulated the dynamic changes of soil water content and salts. The accuracy of the validation period met the requirements. In the validation process, the MRE between the simulated and measured values of soil $NO_3$-N content was between 10.25% and 14.25%; the RMSE was 10.14–11.87 g·kg$^{-1}$; and the $R^2$ was 0.74–0.86. CD increased $NO_3$-N content and crop $NO_3$-N uptake, and reduced $NO_3$-N leaching and loss. We observed a significant difference in soil $NO_3$-N content after irrigation at the budding stage of oilseed sunflower farmland between treatments, with the largest difference seen for the 0–40 cm soil layer. CD improved crop yield. The average oilseed sunflower yields of the CWT1 and CWT2 treatments increased by 4.52% and 3.04% relative to the FD treatment ($p < 0.05$). CD also enhanced nitrogen use efficiency. Thus, in moderately salinized soil, CD of 40 cm (CWT1) is a suitable drainage method in the experimental area.

**Author Contributions:** X.D. and H.S. were involved in designing the manuscript; X.D., H.S. and R.L. analyzed the data and wrote the manuscript; X.D., Q.M., J.Y. and F.T. carried out this experiment. All authors have read and agreed to the published version of the manuscript.

**Funding:** This research was jointly supported by the National Natural Science Foundation of China (51879132, 52269014 and 52009056), the Major Science and Technology Projects of Inner Mongolia (zdzx2018059), the Major Water Conservancy Science and Technology Projects of Inner Mongolia (nsk2018-M5).

**Data Availability Statement:** Data are contained within the article.

**Conflicts of Interest:** The authors declare no conflict of interest.

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
