# Peer review of "Evaluating the Effects of Controlled Drainage on Nitrogen Uptake, Utilization, Leaching, and Loss in Farmland Soil"

_agronomy, doi:10.3390/agronomy13122936_

Round 1

Reviewer 1 Report

Comments and Suggestions for Authors

This drainage study was carried out in a field with a very high silt content, and, therefore, a high leaching rate is expected. In addition to silt content, this soil was later enriched with sand, which increased the water and solute movement in the profile. The impact of this needs to be emphasised more in the discussions and evaluation of the results and comparison of simulations with real data. Especially in this type of soil, nitrogen movement can be modelled very successfully. I recommend that the relationship between measured and predicted values and texture be emphasised in the discussions.

Author Response

Thank you for taking time out of your busy schedule to review the manuscript. Now we have carefully corrected and replied thce manuscript for this revision. The revision instruction are as follows:

Point: This drainage study was carried out in a field with a very high silt content, and, therefore, a high leaching rate is expected. In addition to silt content, this soil was later enriched with sand, which increased the water and solute movement in the profile. The impact of this needs to be emphasised more in the discussions and evaluation of the results and comparison of simulations with real data. Especially in this type of soil, nitrogen movement can be modelled very successfully. I recommend that the relationship between measured and predicted values and texture be emphasised in the discussions.

Response: Thank you very much for the valuable opinions of the reviewer. Your suggestion plays a crucial role in improving the overall and rigorous nature of the article. In order to increase soil permeability in the experiment, we added fine sands (dune sands) to the surface of the soil. This can timely discharge excess water without causing damage. For this impact, corresponding content has been added in the Results and Analysis section and Discussion section of the manuscript. See the line 342-344 and 558-562 in the manuscript.

To sum up, please do not hesitate to contact us if there are any question. Thanks again to reviewers and editors for your hard work! Best wishes to you!

Reviewer 2 Report

Comments and Suggestions for Authors

This manuscript explored the responses of nitrogen uptake, utilization, leaching, and loss in Hetao Irrigation District to subsurface drainage with perforated pipes. The HYDRUS-2D model was also used for simulation to determine the best control drainage mode. The topic is interesting. The manuscript is well written and organized. There are some minor issues need to be addressed before it be accepted for publication in this journal. The specific comments are as following:
1. Abstract:
what is CD and FD here, define as it appeared first time.

2. In the Introduction section, it is suggested to add the advantages of using HYDRUS model in this region.
3. Please provide more details about how the controlled drainage system works.

4. Please describe briefly how the parameters of HYDRUS-2D mode were calibrated and what was the objective function for the calibration.

5. L329: Salts not salinity, please check the whole manuscript.

6. L341: DAS. Please give the full name when it first appeared in the manuscript.

7. L379: the soil water content” should be “NO3-N contents content”.

8. Is it DAS or DOY in the title of Figure 4? Please check it.

Comments on the Quality of English Language

Minor improvement of the English language is needed.

Author Response

Thank you for taking time out of your busy schedule to review the manuscript. Now we have carefully corrected and replied the manuscript for this revision. The  revision instruction are as follows:

his manuscript explored the responses of nitrogen uptake, utilization, leaching, and loss in Hetao Irrigation District to subsurface drainage with perforated pipes. The HYDRUS-2D model was also used for simulation to determine the best control drainage mode. The topic is interesting. The manuscript is well written and organized. There are some minor issues need to be addressed before it be accepted for publication in this journal. The specific comments are as following:

Point 1: Abstract: what is CD and FD here, define as it appeared first time.

Response 1: I am deeply sorry for the confusion caused by the reviewers. I have already defined FD and CD in the manuscript. See the line 10-12 in the manuscript. The specific content is as follows: free drainage (FD) and growth-stage subsurface controlled drainage (CD) at depths of 40 cm(CWT1) and 70 cm (CWT1 and CWT2, respectively).

Point 2: In the Introduction section, it is suggested to add the advantages of using HYDRUS model in this region.

Response 2: Thank you very much for the valuable opinions of the reviewer. Your suggestion plays a crucial role in improving the overall and rigorous nature of the article. I have reorganized the introduction section and added the application advantages of HYDRAS in the region. See the line 89-91 in the manuscript.

Point 3: Please provide more details about how the controlled drainage system works.

Response 3: I am deeply sorry for the confusion caused by the reviewers. According to your request, the specific workflow is described in detail, as shown in the figure below, the height of No. 1, No. 2, and No. 3 drain ports from the ground are 40cm, 70cm, and 100cm respectively. During spring irrigation, the drainage depth is 100cm, close the No.1 and No.2 drainage ports, and open the No.3 drainage port. During the growth stage when the drainage is 40cm, open the No.1 drain and close the No.2 and No.3; during the growth stage when the drainage is 70cm, open the No.2 drain and close the No.1 and No.3 drain; during the growth stage when the drainage is 100cm, open the No.3 drain and close the No.1 and No.2 drain.

Point 4: Please describe briefly how the parameters of HYDRUS-2D mode were calibrated and what was the objective function for the calibration.

Response 4: I am deeply sorry for the confusion caused by the reviewers. Parameters were calibrated using 2020 soil water content and nitrogen content and validated with 2021 data. Model rate-setting and testing were evaluated using mean rela-tive error (MRE), root mean square error (RMSE), and coefficient of determination (R2) for testing. See the line 323-333 in the manuscript.

Point 5: L329: Salts not salinity, please check the whole manuscript.

Response 5: I'm very sorry for the non-standard language that caused the reviewers to have ambiguity. Salinity has been changed to salts. And a comprehensive review and revision of the manuscript was conducted.

Point 6: L341: DAS. Please give the full name when it first appeared in the manuscript.

Response 6: I am deeply sorry for the confusion caused by the reviewers. Defined when first used in the manuscript. See the line 365-366 in the manuscript.

Point 7: L379: “the soil water content” should be “NO3-N content”.

Response 7: I am deeply sorry for the confusion caused by the reviewers. I have made corrections in the manuscript. See the line 381 in the manuscript.

Point 8: Is it DAS or DOY in the title of Figure 4? Please check it.

Response 8: I am deeply sorry for the confusion caused by the reviewers. I have made corrections in the manuscript. See the line 446 in the manuscript.

To sum up, please do not hesitate to contact us if there are any question. Thanks again to reviewers and editors for your hard work! Best wishes to you!

Reviewer 3 Report

Comments and Suggestions for Authors

The authors used the HYDRUS-2D model to simulate the dynamic changes of NO3-N in the 0–100 cm soil layer as well as the NO3-N uptake by crops, leaching after irrigation and fertilization, and loss through subsurface pipes in 2020 (model calibration period) and 2021 (model validation period). The results are of research merit and worthy of publication in journal, but there are some minor issues that need to be addressed.

The first paragraph of the discussion needs to be revised to focus on the results obtained rather than focusing on explaining the feasibility of this model.

The controlled drainage (CD) depths are 40 cm and 70 cm (CWT1 and CWT2 respectively), according to the results of the study, which depth is suitable for crop utilization?

In Figure 4, CD increased crop yield by 4.52% and 3.04% compared with FD treatment (P < 0.05). However, from the results, there was only a 3-4% difference. How to calculate the biostatistical difference?

Comments on the Quality of English Language

Minor editing of English language required

Author Response

Thank you for taking time out of your busy schedule to review the manuscript. Now we have carefully corrected and replied the manuscript for this revision. The  revision instruction are as follows:

The authors used the HYDRUS-2D model to simulate the dynamic changes of NO3-N in the 0–100 cm soil layer as well as the NO3-N uptake by crops, leaching after irrigation and fertilization, and loss through subsurface pipes in 2020 (model calibration period) and 2021 (model validation period). The results are of research merit and worthy of publication in journal, but there are some minor issues that need to be addressed.

Point 1: The first paragraph of the discussion needs to be revised to focus on the results obtained rather than focusing on explaining the feasibility of this model.

Response 1: Thank you very much for the valuable opinions of the reviewer. Your suggestion plays a crucial role in improving the overall and rigorous nature of the article. I have made modifications to the first paragraph of the discussion. I have added the results obtained in the article. See the lines 553-556 and 558-562 in the manuscript.

Point 2: The controlled drainage (CD) depths are 40 cm and 70 cm (CWT1 and CWT2 respectively), according to the results of the study, which depth is suitable for crop utilization?

Response 2: I am deeply sorry for the confusion caused by the reviewers. I have added the corresponding results in the article. See the lines 631-632 in the manuscript. The specific content is as follows:

Thus, in moderately salinized soil, CD of 40 cm (CWT1) is a suitable drainage method in the experimental area.

Point 3: In Figure 4, CD increased crop yield by 4.52% and 3.04% compared with FD treatment (P<0.05). However, from the results, there was only a 3-4% difference. How to calculate the biostatistical difference?

Response 3: In Table 4, CD increased crop yield by 4.52% and 3.04% compared with FD treatment. I used SPSS27.0 software to analyze significance. Firstly, I cleaned and preprocessed the data, and then used the LSD method in the software to analyze the differences in yield between CWT1, CWT2, and FD treatments. The results showed that the differences were statistically significant.

To sum up, please do not hesitate to contact us if there are any question. Thanks again to reviewers and editors for your hard work! Best wishes to you!
